# Constructing a Class of Frozen Jacobian Multi-Step Iterative Solvers for Systems of Nonlinear Equations

**R. H. Al-Obaidi and M. T. Darvishi** *

Department of Mathematics, Faculty of Science, Razi University, Kermanshah 67149, Iran
* Correspondence: darvishi@razi.ac.ir

**Abstract:** In this paper, in order to solve systems of nonlinear equations, a new class of frozen Jacobian multi-step iterative methods is presented. Our proposed algorithms are characterized by a highly convergent order and an excellent efficiency index. The theoretical analysis is presented in detail. Finally, numerical experiments are presented for showing the performance of the proposed methods, when compared with known algorithms taken from the literature.

**Keywords:** iterative method; frozen Jacobian multi-step iterative method; system of nonlinear equations; high-order convergence

## 1. Introduction

Approximating a locally unique solution $\alpha$ of the nonlinear system

$$F(\mathbf{x}) = 0 \tag{1}$$

has many applications in engineering and mathematics [1–4]. In (1), we have $n$ equations with $n$ variables. In fact, $F$ is a vector-valued function with $n$ variables. Several problems arising from the different areas in natural and applied sciences take the form of systems of nonlinear Equation (1) that need to be solved, where $F(\mathbf{x}) = (f_1(\mathbf{x}), f_2(\mathbf{x}), \cdots, f_n(\mathbf{x}))$ such that for all $k = 1, 2, \cdots, n$, $f_k$ is a scalar nonlinear function. Additionally, there are many real life problems for which, in the process of finding their solutions, one needs to solve a system of nonlinear equations, see for example [5–9]. It is known that finding an exact solution $\alpha^t = (\alpha_1, \alpha_2, \cdots, \alpha_n)$ of the nonlinear system (1) is not an easy task, especially when the equation contains terms consisting of logarithms, trigonometric and exponential functions, or a combination of transcendental terms. Hence, in general, one cannot find the solution of Equation (1) analytically, therefore, we have to use iterative methods. Any iterative method starts from one approximation and constructs a sequence such that it converges to the solution of the Equation (1) (for more details, see [10]).

The most commonly used iterative method to solve (1) is the classical Newton method, given by

$$\mathbf{x}^{(k+1)} = \mathbf{x}^{(k)} - J_F(\mathbf{x}^{(k)})^{-1} F(\mathbf{x}^{(k)}),$$

where $J_F(\mathbf{x})$ (or $F'(\mathbf{x})$) is the Jacobian matrix of function $F$, and $\mathbf{x}^{(k)}$ is the $k$-th approximation of the root of (1) with the initial guess $\mathbf{x}^{(0)}$. It is well known that Newton's method is a quadratic convergence method with the efficiency index $\sqrt{2}$ [11]. The third and higher-order methods such as the Halley and Chebyshev methods [12] have little practical value because of the evaluation of the second Frechèt-derivative. However, third and higher-order multi-step methods can be good substitutes because they require the evaluation of the function and its first derivative at different points.

In the recent decades, many authors tried to design iterative procedures with better efficiency and higher order of convergence than the Newton scheme, see, for example, ref. [13–24]

and references therein. However, the accuracy of solutions is highly dependent on the efficiency of the utilized algorithm. Furthermore, at each step of any iterative method, we must find the exact solution of an obtained linear system which is expensive in actual applications, especially when the system size $n$ is very large. However, the proposed higher-order iterative methods are futile unless they have high-order convergence. Therefore, the important aim in developing any new algorithm is to achieve high convergence order with requiring as small as possible the evaluations of functions, derivatives and matrix inversions. Thus, here, we focus on the technique of the frozen Jacobian multi-step iterative algorithms. It is shown that this idea is computationally attractive and economical for constructing iterative solvers because the inversion of the Jacobian matrix (regarding *LU*-decomposition) is performed once. Many researchers have reduced the computational cost of these algorithms by frozen Jacobian multi-step iterative techniques [25–28].

In this work, we construct a new class of frozen Jacobian multi-step iterative methods for solving the nonlinear systems of equations. This is a high-order convergent algorithm with an excellent efficiency index. The theoretical analysis is presented completely. Further, by solving some nonlinear systems, the ability of the methods is compared with some known algorithms.

The rest of this paper is organized as follows. In the following section, we present our new methods with obtaining of their order of convergence. Additionally, their computational efficiency are discussed in general. Some numerical examples are considered in Sections 3 and 4 to show the asymptotic behavior of these methods. Finally, a brief concluding remark is presented in Section 5.

## 2. Constructing New Methods

In this section, two high-order frozen Jacobian multi-step iterative methods to solve systems of nonlinear equations are presented. These come by increasing the convergence in Newton's method and simultaneously decreasing its computational costs. The framework of these Frozen Jacobian multi-step iterative Algorithms (FJA) can be described as

$$
\begin{cases}
\text{No. of steps} & = m > 1, \\
\text{Order of convergence} = m + 1, \\
\text{Function evaluations} & = m, \\
\text{Jacobian evaluations} & = 1, \\
\text{No. of } LU \text{ decomposition} = 1;
\end{cases}
\quad
\text{FJA}:
\begin{cases}
\mathbf{y}_0 = \text{initial guess} \\
\mathbf{y}_1 = \mathbf{y}_0 - J_F(\mathbf{y}_0)^{-1}F(\mathbf{y}_0) \\
\text{for } i = 1 : m - 1 \\
\quad \text{\OE}_i = J_F(\mathbf{y}_0)^{-1}(F(\mathbf{y}_i) + F(\mathbf{y}_{i-1})) \quad (2) \\
\quad \mathbf{y}_{i+1} = \mathbf{y}_{i-1} - \text{\OE}_i \\
\text{end} \\
\mathbf{y}_0 = \mathbf{y}_m.
\end{cases}
$$

In (2), for an $m$-step method ($m > 1$), one needs $m$ function evaluations and only one Jacobian evaluation. Further, the number of *LU* decompositions is one. The order of convergence for such FJA method is $m + 1$. In the right-hand side column of (2), the algorithm is briefy described.

In the following subsections, by choosing two different values for $m$, a third- and a fourth-order frozen Jacobian multi-step iterative algorithm are presented.

### 2.1. The Third-Order FJA

First, we investigate case $m = 2$, that is,

$$
\mathbf{y}^{(k)} = \mathbf{x}^{(k)} - J_F(\mathbf{x}^{(k)})^{-1}F(\mathbf{x}^{(k)}),
$$

$$
\mathbf{x}^{(k+1)} = \mathbf{x}^{(k)} - J_F(\mathbf{x}^{(k)})^{-1}(F(\mathbf{y}^{(k)}) + F(\mathbf{x}^{(k)})),
$$

(3)

we denote this by $M_3$.

### 2.1.1. Convergence Analysis

In this part, we prove that the order of convergence of method (3) is three. First, we need to definition of the Frechèt derivative.

**Definition 1** ([29])**.** *Let F be an operator which maps a Banach space X into a Banach space Y. If there exists a bounded linear operator T from X into Y such that*

$$\lim_{\mathbf{y} \to 0} \frac{\|F(\mathbf{x} + \mathbf{y}) - F(\mathbf{x}) - T(\mathbf{y})\|}{\|\mathbf{y}\|} = 0,$$

*then F is said to be Frechèt differentiable and $F'(\mathbf{x}_0) = T(\mathbf{x}_0)$.*

*For more details on the Frechèt differentiability and Frechèt derivative, we refer the interested readers to a review article by Emmanuel [30] and references therein.*

**Theorem 1.** *Let $F : I \subseteq \mathbb{R}^n \to \mathbb{R}^n$ be a Frechèt differentiable function at each point of an open convex neighborhood I of $\alpha$, the solution of system $F(\mathbf{x}) = 0$. Suppose that $J_F(\mathbf{x}^{(k)})$ is continuous and nonsingular in $\alpha$, then, the sequence $\{\mathbf{x}^{(k)}\}_{(k \geqslant 0)}$ obtained using the iterative method (3) converges to $\alpha$ and its rate of convergence is three.*

**Proof.** Suppose that $E_n = \mathbf{x}^{(n)} - \alpha$, using Taylor's expansion [31], we obtain

$$F(\mathbf{x}^{(n)}) = F(\alpha) + F'(\alpha)E_n + \frac{1}{2!}F''(\alpha)E_n^2 + \frac{1}{3!}F'''(\alpha)E_n^3 + \frac{1}{4!}F''''(\alpha)E_n^4 + \dots$$

as $\alpha$ is the root of $F$ so $F(\alpha) = 0$. As a matter of fact, one may yield the following equations of $F(\mathbf{x}^{(n)})$ and $F'(\mathbf{x}^{(n)})$ in a neighborhood of $\alpha$ by using Taylor's series expansions [32],

$$F(\mathbf{x}^{(n)}) = F'(\alpha)\left[E_n + C_2 E_n^2 + C_3 E_n^3 + C_4 E_n^4 + C_5 E_n^5 + O\|E_n^6\|\right], \tag{4}$$

$$F'(\mathbf{x}^{(n)}) = F'(\alpha)\left[I + 2C_2 E_n + 3C_3 E_n^2 + 4C_4 E_n^3 + 5C_5 E_n^4 + 6C_6 E_n^5 + O\|E_n^6\|\right], \tag{5}$$

wherein $C_n = \frac{[F'(\alpha)]^{-1}F^{(n)}(\alpha)}{n!}$ and $I$ is the identity matrix whose order is the same as the order of the Jacobian matrix. Note that $iC_i E_n^{i-1} \in \mathcal{L}(\mathbb{R}^n)$. Using (4) and (5) we obtain

$$F'(\mathbf{x}^{(n)})^{-1}F(\mathbf{x}^{(n)}) = E_n - C_2 E_n^2 + (2C_2^2 - 2C_3)E_n^3 + (-4C_2^3 + 7C_2 C_3 - 3C_4)E_n^4$$
$$+ (-32C_2^5 + 8C_2^4 - 20C_2^2 C_3 + 10C_2 C_4 + 6C_3^2 - 4C_5)E_n^5 + O\|E_n^6\|.$$

Since $\mathbf{y}^{(n)} = \mathbf{x}^{(n)} - F'(\mathbf{x}^{(n)})^{-1}F(\mathbf{x}^{(n)})$, we find

$$\mathbf{y}^{(n)} = \alpha + C_2 E_n^2 + (-2C_2^2 + 2C_3)E_n^3 + (4C_2^3 - 7C_2 C_3 + 3C_4)E_n^4$$
$$+ (32C_2^5 - 8C_2^4 + 20C_2^2 C_3 - 10C_2 C_4 - 6C_3^2 + 4C_5)E_n^5 + O\|E_n^6\|. \tag{6}$$

By the definition of error term $E_n$, the error term of $\mathbf{y}^{(n)}$ as an approximation of $\alpha$, that is, $\mathbf{y}^{(n)} - \alpha$ is obtained from the second term of the right-hand side of Equation (6). Similarly, the Taylor's expansion of the function $F(\mathbf{y}^{(n)})$ is

$$F(\mathbf{y}^{(n)}) = F'(\alpha)\left[C_2 E_n^2 + (-2C_2^2 + 2C_3)E_n^3 + (5C_2^3 - 7C_2 C_3 + 3C_4)E_n^4 + \right.$$
$$\left. (32C_2^5 - 12C_2^4 + 24C_2^2 - 10C_2 C_4 - 6C_3^2 + 4C_5)E_n^5 + O\|E_n^6\|\right]. \tag{7}$$

From (4) and (7), we obtain

$$(F(\mathbf{x}^{(n)}) + F(\mathbf{y}^{(n)})) = F'(\alpha)\left[E_n + 2C_2 E_n^2 + (-2C_2^2 + 3C_3)E_n^3 + (5C_2^3 - 7C_2C_3 + \right.$$

$$\left. 4C_4)E_n^4 + (32C_2^5 - 12C_2^4 + 24C_2^2 - 10C_2C_4 - 6C_3^2 + 6C_5)E_n^5] + O||E_n^6||\right].$$

Thus,

$$F'(\mathbf{x}^{(n)})^{-1}(F(\mathbf{x}^{(n)}) + F(\mathbf{y}^{(n)})) = E_n - (2C_2^2)E_n^3 + (9C_2^3 - 7C_2C_3)E_n^4$$
$$+(-30C_2^4 + 44C_2^2C_3 - 10C_2C_4 - 6C_3^2 + C_5)E_n^5 + O||E_n^6||.$$

Finally, since

$$\mathbf{x}^{(n+1)} = \mathbf{x}^{(n)} - J_F(\mathbf{x}^{(n)})^{-1}(F(\mathbf{x}^{(n)}) + F(\mathbf{y}^{(n)})),$$

we have

$$\mathbf{x}^{(n+1)} = \alpha - (2C_2^2)E_n^3 - (9C_2^3 - 7C_2C_3)E_n^4 - (-30C_2^4 + 44C_2^2C_3 - 10C_2C_4 + $$
$$\cdots - 6C_3^2 + C_5)E_n^5 + O||E_n^6||. \tag{8}$$

Clearly, the error Equation (8) shows that the order of convergence of the frozen Jacobian multi-step iterative method (3) is three. This completes the proof. $\square$

### 2.1.2. The Computational Efficiency

In this section, we compare the computational efficiency of our third-order scheme (3), denoted as $M_3$, with some existing third-order methods. We will assess the efficiency index of our new frozen Jacobian multi-step iterative method in contrast with the existing methods for systems of nonlinear equations, using two famous efficiency indices. The first one is the classical efficiency index [33] as

$$IE = p^{\frac{1}{c}}$$

where $p$ is the rate of convergence and $c$ stands for the total computational cost per iteration in terms of the number of functional evaluations, such that $c = (rn + mn^2)$ where $r$ refers to the number of function evaluations needed per iteration and $m$ is the number of Jacobian matrix evaluations needed per iteration.

It is well known that the computation of $LU$ factorization by any of the existing methods in the literature normally needs $2n^3/3$ flops in floating point operations, while the floating point operations to solve two triangular systems needs $2n^2$ flops.

The second criterion is the flops-like efficiency index ($FLEI$) which was defined by Montazeri et al. [34] as

$$FLEI = p^{\frac{1}{c}}$$

where $p$ is the order of convergence of the method, $c$ denotes the total computational cost per loop in terms of the number of functional evaluations, as well as the cost of $LU$ factorization for solving two triangular systems (based on the flops).

As the first comparison, we compare $M_3$ with the third-order method given by Darvishi [35], which is denoted as $M_{3,1}$

$$\mathbf{y}^{(k)} = \mathbf{x}^{(k)} - J_F(\mathbf{x}^{(k)})^{-1}F(\mathbf{x}^{(k)}),$$
$$\mathbf{x}^{(k+1)} = \mathbf{x}^{(k)} - 2(J_F(\mathbf{x}^{(k)}) + J_F(\mathbf{y}^{(k)}))^{-1}F(\mathbf{x}^{(k)}).$$

The second iterative method shown by $M_{3,2}$ is the following third-order method introduced by Hernández [36]

$$\mathbf{y}^{(k)} = \mathbf{x}^{(k)} - \tfrac{1}{2}J_F(\mathbf{x}^{(k)})^{-1}F(\mathbf{x}^{(k)}),$$
$$\mathbf{x}^{(k+1)} = \mathbf{x}^{(k)} + J_F(\mathbf{x}^{(k)})^{-1}(J_F(\mathbf{y}^{(k)}) - 2J_F(\mathbf{x}^{(k)})) \times J_F(\mathbf{x}^{(k)})^{-1}F(\mathbf{x}^{(k)}).$$

Another method is the following third-order iterative method given by Babajee et al. [37], $M_{3,3}$,

$$\mathbf{y}^{(k)} = \mathbf{x}^{(k)} - J_F(\mathbf{x}^{(k)})^{-1}F(\mathbf{x}^{(k)}),$$
$$\mathbf{x}^{(k+1)} = \mathbf{x}^{(k)} + \tfrac{1}{2}J_F(\mathbf{x}^{(k)})^{-1}(J_F(\mathbf{y}^{(k)}) - 3J_F(\mathbf{x}^{(k)})) \times J_F(\mathbf{x}^{(k)})^{-1}F(\mathbf{x}^{(k)}).$$

Finally, the following third-order iterative method, $M_{3,4}$, ref. [38] is considered

$$\mathbf{y}^{(k)} = \mathbf{x}^{(k)} - \tfrac{2}{3}J_F(\mathbf{x}^{(k)})^{-1}F(\mathbf{x}^{(k)}),$$
$$\mathbf{x}^{(k+1)} = \mathbf{x}^{(k)} - 4(J_F(\mathbf{x}^{(k)}) + 3J_F(\mathbf{y}^{(k)}))^{-1}F(\mathbf{x}^{(k)}).$$

The computational efficiency of our third-order method revealed that our method, $M_3$, is the best one in respect with methods $M_{3,1}$, $M_{3,2}$, $M_{3,3}$ and $M_{3,4}$, as presented in Table 1, and Figures 1 and 2.

**Table 1.** Comparison of efficiency indices between $M_3$ and other third-order methods.

| Methods | $M_3$ | $M_{3,1}$ | $M_{3,2}$ | $M_{3,3}$ | $M_{3,4}$ |
|---|---|---|---|---|---|
| No. of steps | 2 | 2 | 2 | 2 | 2 |
| Order of convergence | 3 | 3 | 3 | 3 | 3 |
| Functional evaluations | $2n + n^2$ | $n + 2n^2$ | $n + 2n^2$ | $n + 2n^2$ | $n + 2n^2$ |
| The classical efficiency index (IE) | $3^{1/(2n+n^2)}$ | $3^{1/(n+2n^2)}$ | $3^{1/(n+2n^2)}$ | $3^{1/(n+2n^2)}$ | $3^{1/(n+2n^2)}$ |
| No. of *LU* decompositions | 1 | 2 | 1 | 1 | 2 |
| Cost of *LU* decompositions | $\frac{2n^3}{3}$ | $\frac{4n^3}{3}$ | $\frac{2n^3}{3}$ | $\frac{2n^3}{3}$ | $\frac{4n^3}{3}$ |
| Cost of linear systems (based on flops) | $\frac{2n^3}{3} + 4n^2$ | $\frac{4n^3}{3} + 4n^2$ | $\frac{5n^3}{3} + 2n^2$ | $\frac{5n^3}{3} + 2n^2$ | $\frac{4n^3}{3} + 4n^2$ |
| Flops-like efficiency index (FLEI) | $3^{1/(\frac{2n^3}{3}+5n^2+2n)}$ | $3^{1/(\frac{4n^3}{3}+6n^2+n)}$ | $3^{1/(\frac{5n^3}{3}+4n^2+n)}$ | $3^{1/(\frac{5n^3}{3}+4n^2+n)}$ | $3^{1/(\frac{4n^3}{3}+6n^2+n)}$ |

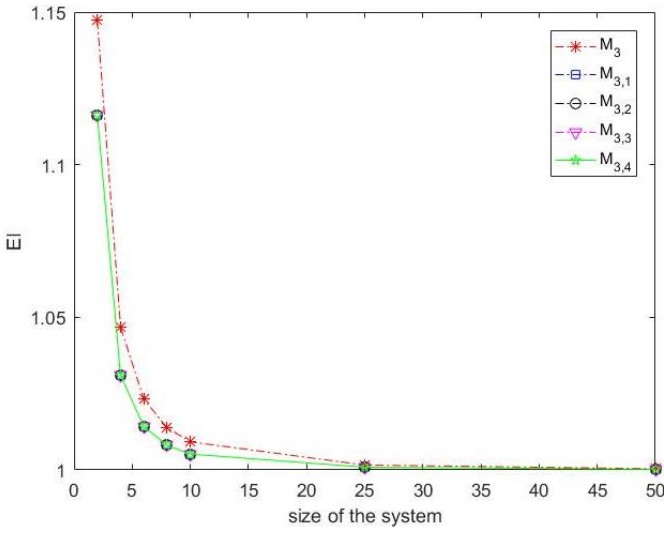

**Figure 1.** The classical efficiency index for methods $M_3$, $M_{3,1}$, $M_{3,2}$, $M_{3,3}$ and $M_{3,4}$.

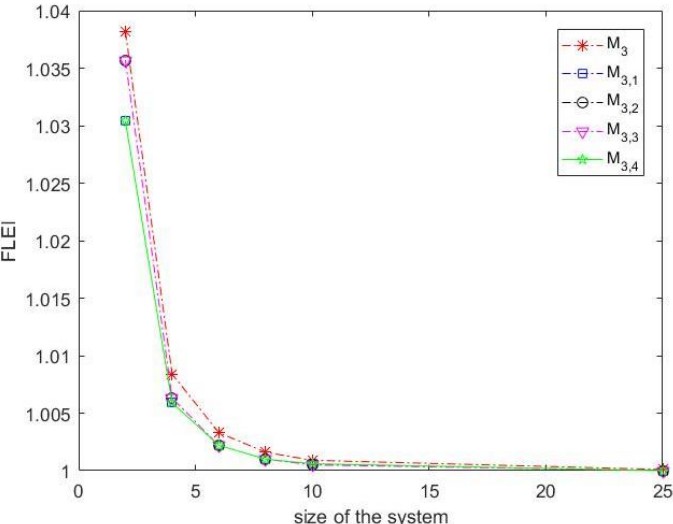

**Figure 2.** The flops-like efficiency index for methods $M_3$, $M_{3,1}$, $M_{3,2}$, $M_{3,3}$ and $M_{3,4}$.

### 2.2. The Fourth-Order FJA

By setting $m = 3$ in FJA, the following three-step algorithm is deduced

$$
\begin{aligned}
\mathbf{y}^{(k)} &= \mathbf{x}^{(k)} - J_F(\mathbf{x}^{(k)})^{-1} F(\mathbf{x}^{(k)}), \\
\mathbf{z}^{(k)} &= \mathbf{x}^{(k)} - J_F(\mathbf{x}^{(k)})^{-1} (F(\mathbf{y}^{(k)}) + F(\mathbf{x}^{(k)})), \\
\mathbf{x}^{(k+1)} &= \mathbf{y}^{(k)} - J_F(\mathbf{x}^{(k)})^{-1} (F(\mathbf{z}^{(k)}) + F(\mathbf{y}^{(k)})).
\end{aligned}
\tag{9}
$$

In the following subsections, the order of convergence and efficiency indices are obtained for the method described in (9).

#### 2.2.1. Convergence Analysis

The frozen Jacobian three-step iterative process (9) has the rate of convergence order four by using three evaluations of function $F$ and one first-order Frechèt derivative $F$ per full iterations. To avoid any repetition, we take a sketch of proof on this subject. Similar to the proof of Theorem 1, by setting $\mathbf{z}^{(k)} = \mathbf{x}^{(k+1)}$ in (8) we obtain

$$
\begin{aligned}
F(\mathbf{z}^{(k)}) = F'(\alpha)[2C_2^2 E_n^3 &+ (-9C_2^3 + 7C_2 C_3)E_n^4 + (30C_2^4 - 44C_2^2 C_3 + \\
&\dots + 10C_2 C_4 - C_5)E_n^5 + O||E_n^6||].
\end{aligned}
$$

Hence,

$$
\begin{aligned}
(F(\mathbf{z}^{(k)}) + F(\mathbf{y}^{(k)})) = F'(\alpha)\bigg[ C_2 E_n^2 &+ 2C_3 E_n^3 + (-4C_2^3 + 3C_4)E_n^4 \\
&+ (32C_2^5 + 18C_2^4 - 20C_2^2 C_3 + 3C_5)E_n^5 + O||E_n^6|| \bigg].
\end{aligned}
\tag{10}
$$

Therefore, from (5) and (10), we find

$$
\begin{aligned}
F'(\mathbf{x}^{(k)})^{-1}(F(\mathbf{z}^{(k)}) + F(\mathbf{y}^{(k)})) = \bigg[ C_2 E_n^2 &+ (-2C_2^2 + 2C_3)E_n^3 + (-7C_2 C_3 + \dots \\
&+ 3C_4)E_n^4 + (18C_2^4 - 10C_2 C_4 - 6C_3^2 + 3C_5)E_n^5 + O||E_n^6|| \bigg].
\end{aligned}
\tag{11}
$$

Since we have $\mathbf{x}^{(k+1)} = \mathbf{y}^{(k)} - J_F(\mathbf{x}^{(k)}))^{-1}(F(\mathbf{z}^{(k)}) + F(\mathbf{y}^{(k)}))$ from (6) and (11), the following result is obtained

$$\mathbf{x}^{(k+1)} = \alpha + (4C_2^3)E_n^4 + (32C_2^5 - 26C_2^4 + 20C_2^2C_3 + C_5)E_n^5 + O||E_n^6||. \tag{12}$$

This completes the proof, since error Equation (12) shows that the order of convergence of the frozen Jacobian multi-step iterative method (9) is four.

### 2.2.2. The Computational of Efficiency

Now, we compare the computational efficiency of our fourth-order scheme (9), called by $M_4$, with some existing fourth-order methods. The considered methods are: the third-order method $M_{4,1}$ given by Sharma et al. [39],

$$\begin{aligned}
\mathbf{y}^{(k)} &= \tfrac{2}{3}\mathbf{x}^{(k)} - J_F(\mathbf{x}^{(k)})^{-1}F(\mathbf{x}^{(k)}), \\
\mathbf{x}^{(k+1)} &= \mathbf{x}^{(k)} - \tfrac{1}{2}\left[ -I + \tfrac{9}{4}J_F(\mathbf{y}^{(k)})^{-1}J_F(\mathbf{x}^{(k)}) + \tfrac{3}{4}J_F(\mathbf{x}^{(k)})^{-1}J_F(\mathbf{y}^{(k)}) \right] \\
&\quad \times J_F(\mathbf{x}^{(k)})^{-1}F(\mathbf{x}^{(k)}),
\end{aligned}$$

the fourth-order iterative method $M_{4,2}$ given by Darvishi and Barati [40],

$$\begin{aligned}
\mathbf{y}^{(k)} &= \mathbf{x}^{(k)} - J_F(\mathbf{x}^{(k)})^{-1}F(\mathbf{x}^{(k)}), \\
\mathbf{z}^{(k)} &= \mathbf{x}^{(k)} - J_F(\mathbf{x}^{(k)})^{-1}\left( F(\mathbf{y}^{(k)}) + F(\mathbf{x}^{(k)}) \right), \\
\mathbf{x}^{(k+1)} &= \mathbf{x}^{(k)} - \left[ \tfrac{1}{6}J_F(\mathbf{x}^{(k)}) + \tfrac{2}{3}J_F(\tfrac{(\mathbf{x}^{(k)}+\mathbf{z}^{(k)})}{2}) + \tfrac{1}{6}J_F(\mathbf{z}^{(k)}) \right]^{-1}F(\mathbf{x}^{(k)}),
\end{aligned}$$

the fourth-order iterative method $M_{4,3}$ given by Soleymani et al. [34,41],

$$\begin{aligned}
\mathbf{y}^{(k)} &= \tfrac{2}{3}\mathbf{x}^{(k)} - J_F(\mathbf{x}^{(k)})^{-1}F(\mathbf{x}^{(k)}), \\
\mathbf{x}^{(k+1)} &= \mathbf{x}^{(k)} - \left[ I - \tfrac{3}{8}\left( I - (J_F(\mathbf{y}^{(k)})^{-1}J_F(\mathbf{x}^{(k)}))^2 \right) \right] J_F(\mathbf{x}^{(k)})^{-1}F(\mathbf{x}^{(k)}),
\end{aligned}$$

and the following Jarratt fourth-order method $M_{4,4}$ [42],

$$\begin{aligned}
\mathbf{y}^{(k)} &= \tfrac{2}{3}\mathbf{x}^{(k)} - J_F(\mathbf{x}^{(k)})^{-1}F(\mathbf{x}^{(k)}), \\
\mathbf{x}^{(k+1)} &= \mathbf{x}^{(k)} - \tfrac{1}{2}\left( 3J_F(\mathbf{y}^{(k)}) - J_F(\mathbf{x}^{(k)}) \right)^{-1}\left( 3J_F(\mathbf{y}^{(k)}) + J_F(\mathbf{x}^{(k)}) \right) \\
&\quad \times J_F(\mathbf{x}^{(k)})^{-1}F(\mathbf{x}^{(k)}).
\end{aligned}$$

The computational efficiency of our fourth-order method showed that our method $M_4$ is better than methods $M_{4,1}$, $M_{4,2}$, $M_{4,3}$ and $M_{4,4}$ as the comparison results are presented in Table 2, and Figures 3 and 4. As we can see from Table 2, the indices of our method $M_4$ are better than similar ones in methods $M_{4,1}$, $M_{4,2}$, $M_{4,3}$ and $M_{4,4}$. Furthermore, Figures 3 and 4 show the superiority of our method in respect with the another schemes.

**Table 2.** Comparison of efficiency indices between $M_4$ and other fourth-order methods.

| Methods | $M_4$ | $M_{4,1}$ | $M_{4,2}$ | $M_{4,3}$ | $M_{4,4}$ |
|---|---|---|---|---|---|
| No. of steps | 3 | 2 | 3 | 2 | 2 |
| Order of convergence | 4 | 4 | 4 | 4 | 4 |
| Functional evaluations | $3n + n^2$ | $n + 2n^2$ | $2n + 3n^2$ | $n + 2n^2$ | $n + 2n^2$ |
| The classical efficiency index (IE) | $4^{1/(3n+n^2)}$ | $4^{1/(n+2n^2)}$ | $4^{1/(2n+3n^2)}$ | $4^{1/(n+2n^2)}$ | $4^{1/(n+2n^2)}$ |
| No. of $LU$ decompositions | 1 | 2 | 2 | 2 | 2 |
| Cost of $LU$ decompositions | $\frac{2n^3}{3}$ | $\frac{4n^3}{3}$ | $\frac{4n^3}{3}$ | $\frac{4n^3}{3}$ | $\frac{4n^3}{3}$ |
| Cost of linear systems (based on flops) | $\frac{2n^3}{3} + 6n^2$ | $\frac{10n^3}{3} + 2n^2$ | $\frac{4n^3}{3} + 6n^2$ | $\frac{7n^3}{3} + 2n^2$ | $7\frac{n^3}{3} + 2n^2$ |
| Flops-like efficiency index (FLEI) | $4^{1/(\frac{2n^3}{3}+7n^2+3n)}$ | $4^{1/(\frac{10n^3}{3}+4n^2+n)}$ | $4^{1/(\frac{4n^3}{3}+9n^2+2n)}$ | $4^{1/(\frac{7n^3}{3}+4n^2+n)}$ | $4^{1/(\frac{7n^3}{3}+4n^2+n)}$ |

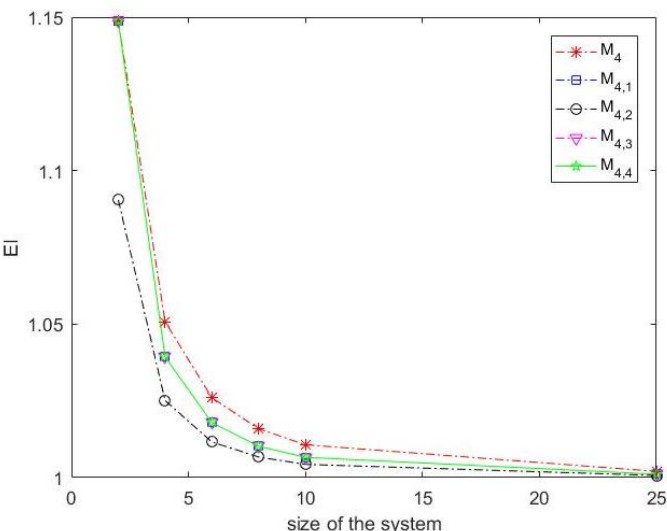

**Figure 3.** The classical efficiency index for methods $M_4$, $M_{4,1}$, $M_{4,2}$, $M_{4,3}$ and $M_{4,4}$.

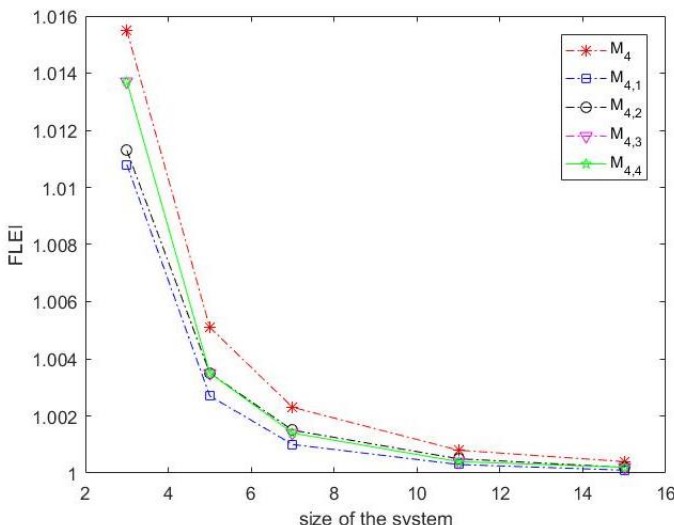

**Figure 4.** The Flops-like efficiency index for methods $M_4$, $M_{4,1}$, $M_{4,2}$, $M_{4,3}$ and $M_{4,4}$.

## 3. Numerical Results

In order to check the validity and efficiency of our proposed frozen Jacobian multi-step iterative methods, three test problems are considered to illustrate convergence and computation behaviors such as efficiency index and some another indices of the frozen Jacobian multi-step iterative methods. Numerical computations have been performed using variable precision arithmetic that uses floating point representation of 100 decimal digits of mantissa in MATLAB. The computer specifications are: Intel(R) Core(TM) i7-1065G7 CPU 1.30 GHz with 16.00 GB of RAM on Windows 10 pro.

**Experiment 1.** We begin with the following nonlinear system of $n$ equations [43],

$$f_i(\mathbf{x}) = \cos(x_i) - 1, \quad i = 1, 2, \dots, n. \tag{13}$$

The exact zero of $F(\mathbf{x}) = (f_1(\mathbf{x}), f_2(\mathbf{x}), \dots, f_n(\mathbf{x}))^t = 0$ is $(0, 0, \dots, 0)^t$. To solve (13), we set the initial guess as $(0.78, 0.78, \dots, 0.78)^t$. The stopping criterion is selected as $||f(\mathbf{x}^{(k)})|| \leq 10^{-3}$.

**Experiment 2.** The next test problem is the following system of nonlinear equations [44],

$$f_i(\mathbf{x}) = (1 - x_i^2) + x_i(1 + x_i x_{n-2} x_{n-1} x_n) - 2, \quad i = 1, 2, \ldots, n. \tag{14}$$

The exact root of $F(\mathbf{x}) = 0$ is $(1, 1, \ldots, 1)^t$. To solve (14), the initial guess is taken as $(2, 2, \ldots, 2)^t$. The stopping criterion is selected as $||f(\mathbf{x}^{(k)})|| \leq 10^{-8}$.

**Experiment 3.** The last test problem is the following nonlinear system [9],

$$\begin{aligned} f_i(\mathbf{x}) &= x_i^2 x_{i+1} - 1, \quad i = 1, 2, \ldots, n-1, \\ f_n(\mathbf{x}) &= x_n^2 x_1 - 1, \end{aligned} \tag{15}$$

with the exact solution $(1, 1, \ldots, 1)^t$. To solve (15), the initial guess and the stopping criterion are respectively considered as $(3, 3, \ldots, 3)^t$ and $||f(\mathbf{x}^{(k)})|| \leq 10^{-8}$.

Table 3 shows the comparison results between our third-order frozen Jacobian two-step iterative method $M_3$ and some third-order frozen Jacobian iterative methods, namely, $M_{3,1}$, $M_{3,2}$, $M_{3,3}$ and $M_{3,4}$. For all test problems, two different values for $n$ are considered, namely, $n = 50, 100$. As this table shows, in all cases, our method works better than the others. Similarly, in Table 4, CPU time and number of iterations are presented for our fourth-order method, namely, $M_4$ and methods $M_{4,1}$, $M_{4,2}$, $M_{4,3}$ and $M_{4,4}$. Similar to $M_3$, the CPU time for $M_4$ is less than the CPU time for the other methods. These tables show superiority of our methods in respect with the other ones. In Tables 3 and 4, *it* shows the number of iterations.

**Table 3.** Comparison results between $M_3$ and other third-order methods.

| Methods | Experiment 1 | | | Experiment 2 | | | Experiment 3 | | |
|---|---|---|---|---|---|---|---|---|---|
| | *n* | *it* | cpu | *n* | *it* | cpu | *n* | *it* | cpu |
| $M_3$ | 50 | 4 | 7.7344 | 50 | 5 | 10.6250 | 50 | 5 | 10.4844 |
| | 100 | 5 | 59.6406 | 100 | 5 | 59.8594 | 100 | 5 | 60.0313 |
| $M_{3,1}$ | 50 | 4 | 11.0625 | 50 | 5 | 13.8125 | 50 | 5 | 14.1406 |
| | 100 | 4 | 69.4219 | 100 | 5 | 87.3594 | 100 | 5 | 87.4063 |
| $M_{3,2}$ | 50 | 4 | 18.7188 | 50 | 5 | 24.9375 | 50 | 5 | 21.5469 |
| | 100 | 5 | 157.2344 | 100 | 5 | 143.7344 | 100 | 5 | 146.2656 |
| $M_{3,3}$ | 50 | 4 | 20.7031 | 50 | 5 | 23.1563 | 50 | 5 | 24.2969 |
| | 100 | 5 | 153.1719 | 100 | 5 | 143.2969 | 100 | 5 | 145.4063 |
| $M_{3,4}$ | 50 | 4 | 13.1719 | 50 | 5 | 13.2500 | 50 | 4 | 11.0156 |
| | 100 | 4 | 73.2500 | 100 | 5 | 88.2031 | 100 | 4 | 70.2500 |

**Table 4.** Comparison results between $M_4$ and other fourth-order methods.

| Methods | Experiment 1 | | | Experiment 2 | | | Experiment 3 | | |
|---|---|---|---|---|---|---|---|---|---|
| | *n* | *it* | cpu | *n* | *it* | cpu | *n* | *it* | cpu |
| $M_4$ | 50 | 4 | 12.2463 | 50 | 4 | 13.3218 | 50 | 4 | 11.5781 |
| | 100 | 4 | 78.1563 | 100 | 5 | 94.9063 | 100 | 4 | 74.2969 |
| $M_{4,1}$ | 50 | 4 | 23.6875 | 50 | 4 | 21.9531 | 50 | 4 | 21.7969 |
| | 100 | 4 | 151.9844 | 100 | 4 | 144.7656 | 100 | 4 | 140.8438 |
| $M_{4,2}$ | 50 | 3 | 15.3906 | 50 | 4 | 18.9531 | 50 | 4 | 18.6875 |
| | 100 | 4 | 121.6563 | 100 | 4 | 122.7344 | 100 | 4 | 118.5781 |
| $M_{4,3}$ | 50 | 3 | 12.2188 | 50 | 4 | 17.8750 | 50 | 4 | 15.2656 |
| | 100 | 4 | 97.5469 | 100 | 4 | 99.0469 | 100 | 4 | 97.1250 |
| $M_{4,4}$ | 50 | 3 | 16.4688 | 50 | 4 | 21.7344 | 50 | 4 | 20.7188 |
| | 100 | 3 | 109.1719 | 100 | 4 | 152.0156 | 100 | 4 | 140.2969 |

### 4. Another Comparison

In the previous parts, we presented some comparison results between our methods $M_3$ and $M_4$ with some another frozen Jacobian multi-step iterative methods from third- and fourth-order methods. In this section, we compare our presented methods with three other methods which are fourth- and fifth-order ones. As Tables 5 and 6 and Figures 5 and 6 show, our methods are also better than these methods.

**First.** The fourth-order method given by Qasim et al. [25], $M_A$,

$$
\begin{aligned}
J_F(\mathbf{x}^{(k)})\theta_1 &= F(\mathbf{x}^{(k)}), \\
\mathbf{y}^{(k)} &= \mathbf{x}^{(k)} - \theta_1, \\
J_F(\mathbf{x}^{(k)})\theta_2 &= F(\mathbf{y}^{(k)}), \\
J_F(\mathbf{x}^{(k)})\theta_3 &= J_F(\mathbf{y}^{(k)})\theta_2, \\
\mathbf{x}^{(k+1)} &= \mathbf{y}^{(k)} - 2\theta_2 + \theta_3.
\end{aligned}
$$

**Second.** The fourth-order Newton-like method by Amat et al. [26], $M_B$,

$$
\begin{aligned}
\mathbf{y}^{(k)} &= \mathbf{x}^{(k)} - J_F(\mathbf{x}^{(k)})^{-1}F(\mathbf{x}^{(k)}), \\
\mathbf{z}^{(k)} &= \mathbf{y}^{(k)} - J_F(\mathbf{x}^{(k)})^{-1}F(\mathbf{y}^{(k)}), \\
\mathbf{x}^{(k+1)} &= \mathbf{z}^{(k)} - J_F(\mathbf{x}^{(k)})^{-1}F(\mathbf{z}^{(k)}).
\end{aligned}
$$

**Third.** The fifth-order iterative method by Ahmad et al. [28], $M_C$,

$$
\begin{aligned}
J_F(\mathbf{x}^{(k)})\theta_1 &= F(\mathbf{x}^{(k)}), \\
\mathbf{y}^{(k)} &= \mathbf{x}^{(k)} - \theta_1, \\
J_F(\mathbf{x}^{(k)})\theta_2 &= F(\mathbf{y}^{(k)}), \\
\mathbf{z}^{(k)} &= \mathbf{y}^{(k)} - 3\theta_2, \\
J_F(\mathbf{x}^{(k)})\theta_3 &= J_F(\mathbf{z}^{(k)})\theta_2, \\
J_F(\mathbf{x}^{(k)})\theta_4 &= J_F(\mathbf{z}^{(k)})\theta_3, \\
\mathbf{x}^{(k+1)} &= \mathbf{y}^{(k)} - \tfrac{7}{4}\theta_2 + \tfrac{1}{2}\theta_3 + \tfrac{1}{4}\theta_4.
\end{aligned}
$$

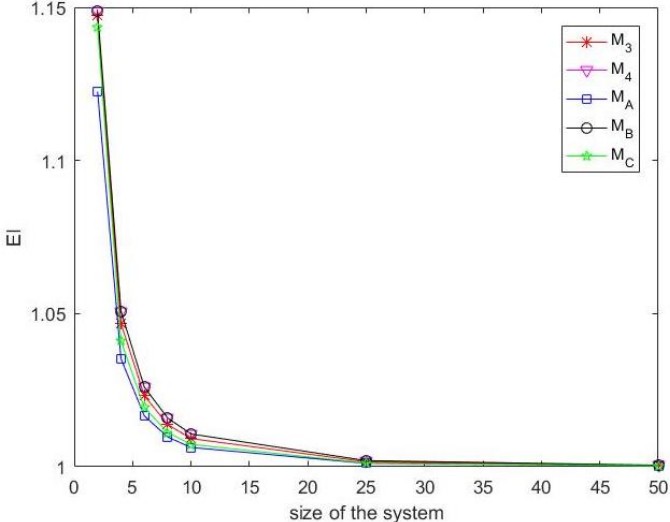

**Figure 5.** The classical efficiency index for $M_3$, $M_4$, $M_A$, $M_B$ and $M_C$.

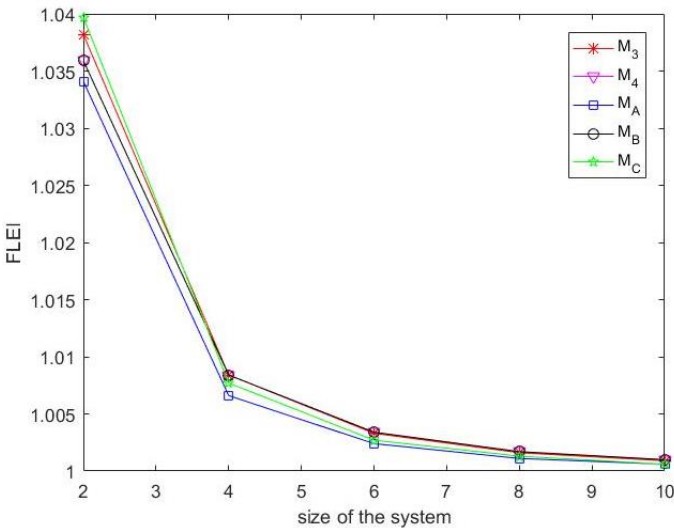

**Figure 6.** The Flops-like efficiency index for $M_3$, $M_4$, $M_A$, $M_B$ and $M_C$.

**Table 5.** Numerical results for comparing of $M_3$ and $M_4$ with $M_A$, $M_B$ and $M_C$.

| Methods | $M_3$ | $M_4$ | $M_A$ | $M_B$ | $M_C$ |
|---|---|---|---|---|---|
| No. of steps | 2 | 3 | 2 | 3 | 3 |
| Order of convergence | 3 | 4 | 4 | 4 | 5 |
| Functional evaluations | $2n + n^2$ | $3n + n^2$ | $2n + 2n^2$ | $3n + n^2$ | $2n + 2n^2$ |
| The classical efficiency index (IE) | $3^{1/(2n+n^2)}$ | $4^{1/(3n+n^2)}$ | $4^{1/(2n+2n^2)}$ | $4^{1/(3n+n^2)}$ | $5^{1/(2n+2n^2)}$ |
| No. of $LU$ decompositions | 1 | 1 | 1 | 1 | 1 |
| Cost of $LU$ decompositions | $\frac{2n^3}{3}$ | $\frac{2n^3}{3}$ | $\frac{2n^3}{3}$ | $\frac{2n^3}{3}$ | $\frac{2n^3}{3}$ |
| Cost of linear systems (based on flops) | $\frac{2n^3}{3} + 4n^2$ | $\frac{2n^3}{3} + 6n^2$ | $\frac{5n^3}{3} + 4n^2$ | $\frac{2n^3}{3} + 6n^2$ | $\frac{5n^3}{3} + 4n^2$ |
| Flops-like efficiency index (FLEI) | $3^{1/(\frac{2n^3}{3}+5n^2+2n)}$ | $4^{1/(\frac{2n^3}{3}+7n^2+3n)}$ | $4^{1/(\frac{5n^3}{3}+6n^2+2n)}$ | $4^{1/(\frac{2n^3}{3}+7n^2+3n)}$ | $5^{1/(\frac{5n^3}{3}+6n^2+2n)}$ |

The comparison results of computational efficiency between our methods $M_3$ and $M_4$ with selected methods $M_A$, $M_B$ and $M_C$ are presented in Table 5. Additionally, Figures 5 and 6 show the graphical comparisons between these methods. Finally, Table 6 shows CPU time and number of iterations to solve our test problems by methods $M_3$, $M_4$, $M_A$, $M_B$ and $M_C$. These numerical and graphical reports show the quality of our algorithms.

**Table 6.** Comparison results between $M_3$, $M_4$, $M_A$, $M_B$ and $M_C$.

| Methods | Experiment 1 | | | Experiment 2 | | | Experiment 3 | | |
|---|---|---|---|---|---|---|---|---|---|
| | $n$ | $it$ | CPU | $n$ | $it$ | CPU | $n$ | $it$ | CPU |
| $M_3$ | 50 | 4 | 7.7344 | 50 | 5 | 10.6250 | 50 | 5 | 10.4844 |
| | 100 | 5 | 59.6406 | 100 | 5 | 59.8594 | 100 | 5 | 60.0313 |
| $M_4$ | 50 | 4 | 12.2463 | 50 | 4 | 13.3218 | 50 | 4 | 11.5781 |
| | 100 | 4 | 78.1563 | 100 | 5 | 94.9063 | 100 | 4 | 74.2969 |
| $M_A$ | 50 | 6 | 23.1875 | 50 | 7 | 25.0625 | 50 | 6 | 25.4063 |
| | 100 | 6 | 139.5625 | 100 | 7 | 173.8125 | 100 | 6 | 150.8594 |
| $M_B$ | 50 | 4 | 15.2509 | 50 | 4 | 12.1563 | 50 | 4 | 12.9219 |
| | 100 | 4 | 76.1406 | 100 | 5 | 91.1719 | 100 | 4 | 71.6406 |
| $M_C$ | 50 | 4 | 23.4688 | 50 | 4 | 23.4854 | 50 | 4 | 22.1531 |
| | 100 | 4 | 139.9844 | 100 | 4 | 185.1406 | 100 | 4 | 138.4063 |



## 5. Conclusions

In this article, two new frozen Jacobian two- and three-step iterative methods to solve systems of nonlinear equations are presented. For the first method, we proved that the order of convergence is three, while for the second one, a fourth-order convergence is proved. By solving three different examples, one may see our methods work as well. Further, the CPU time of our methods is less than some selected frozen Jacobian multi-step iterative methods in the literature. Moreover, other indices of our methods such as number of steps, functional evaluations, the classical efficiency index, and so on, are better than these indices for other methods. This class of the frozen Jacobian multi-step iterative methods can be a pattern for new research on the frozen Jacobian iterative algorithms.

**Author Contributions:** Investigation, R.H.A.-O. and M.T.D.; Project administration, M.T.D.; Resources, R.H.A.-O.; Supervision, M.T.D.; Writing—original draft, M.T.D. All authors have read and agreed to the published version of the manuscript.

**Funding:** This research received no external funding.

**Institutional Review Board Statement:** Not applicable.

**Informed Consent Statement:** Not applicable.

**Data Availability Statement:** Not applicable.

**Acknowledgments:** The authors would like to thank the editor of the journal and three anonymous reviewers for their generous time in providing detailed comments and suggestions that helped us to improve the paper.

**Conflicts of Interest:** The authors declare no conflict of interest.

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
