# Peer review of "Constructing a Class of Frozen Jacobian Multi-Step Iterative Solvers for Systems of Nonlinear Equations"

_mathematics, doi:10.3390/math10162952_

Round 1
Reviewer 1 Report
The manuscript under consideration has enough mathematical content and the derivations are correct. However, three drawbacks are present.
1) The language is often poor, sometimes it is difficult to understand
2) Some general references for motivating the work are missing
3) The numerical experiments do not compare with algorithms of the same type
As first example of 1), I write the Abstract again and I give few suggestions to the rest, but the language should be checked in all the manuscript.
In this paper, order to solve systems of nonlinear equations, a new class of frozen Jacobian iterative methods is presented. Our proposed algorithms are characterized by a highly convergent order and an excellent efficiency index. The theoretical analysis is presented in detail. Finally numerical experiments are presented for showing the performance of the proposed methods, when compared with known algorithms taken from the literature.
Page 1, lines 14-15: the main verb is missing.
Page 1, line 15: at the end of “many applications in engineering and mathematics” please add relevant references.
Page 1, line 19: please add “scalar” between “is a” and “nonlinear”.
Page 1, line 20: at the end of “in many areas” please add relevant references.
Page 1, line 23: please delete “any”.
Page 3, line 88: please replace “(6) we have” with “(6), we obtain”
Page 3, line 89: please replace “hence we have” with “, we find”
Page 3, line 90: please replace “we get” with “we deduce”
Page 4, line 91: please replace “(9) we get” with “(9), we obtain”
Page 4: the part between “two triangular systems” and “is a matrix” has to be rewritten since it is not understandable.
Page 6, line 12: please replace “for method” with “for the method described in”
Page 6, line 129: please replace “Therefore from (19) and (6) we can get” with “Therefore, from (19) and (6), we find”
Page 7, line 132: please replace “proof. Because error equation” with “proof, because the error equation in”
Page 7, line 133: please add “the” between “of” and “frozen”
Rearding the numerical experiments the authors compare their methods with thet in [13,24,25,26] which are published in the range 2007-2012, while they ignore the methods in [11,12,13,14] which are more recent and of the same type as the techniques considered in the present manuscript. I invite to make a more complete comparison.
In conclusion I ask for a major revision.
Author Response
The file is attached.

Reviewer 2 Report
This paper is concerned with a new class of frozen Jacobian iterative methods for the nonlinear System of equations. This is a high order convergent algorithm with an excellent efficiency index. The theoretical analysis is presented completely. Further, by solving some nonlinear systems, the ability of the methods is compared with some known algorithms. The authors may be advised to do the following minor revisions before its publication in MDPI (Mathematics):
1. Section 2, line 85: How to deduce the expression of derivative of F, i.e., Eq. (6)?
2. Section 2, line 90: How to obtain the expression of Eq. (9)? Whether through Taylor series?
3. The names "frozen multi step iterative method" and "frozen Jacobian multi-step iterative method" appear alternately and are not unified in the whole paper.
4. The literature survey is fine. In the paper, you discuss the multi-step iterative method. Add some references regarding the methods similar to multi-step iterative method (e.g., homotopy method, multigrid method):
1. Ullah, M. Z., Serra-Capizzano, S., & Ahmad, F. (2015). An efficient multi-step iterative method for computing the numerical solution of systems of nonlinear equations associated with ODEs. Applied Mathematics and Computation, 250, 249-259.
2. Pacurar, M. (2009). Approximating common fixed points of Prešic-Kannan type operators by a multi-step iterative method. An. St. Univ. Ovidius Constanta, 17(1), 153-168.
3. Rafiq, A., & Rafiullah, M. (2009). Some multi-step iterative methods for solving nonlinear equations. Computers & Mathematics with Applications, 58(8), 1589-1597.
4. Aremu, K. O., Izuchukwu, C., Ogwo, G. N., & Mewomo, O. T. (2021). Multi-step Iterative algorithm for minimization and fixed point problems in p-uniformly convex metric spaces. Journal of industrial & management optimization, 17(4), 2161.
5. Soleymani, F., Lotfi, T., & Bakhtiari, P. (2014). A multi-step class of iterative methods for nonlinear systems. Optimization Letters, 8(3), 1001-1015.
Author Response
The file is attached.

Reviewer 3 Report
There are a lot of errors in typos and grammar. For example,
(18) hould be changed into $F(x)= (f_1 (x), f_x (x) , \cdots , f_n (x))$
(29) explain $x^{(k)}$
(64) Eq. (3) is not understable
(68) , that is,
(72) What is the definition of ` sufficiently Frechet differentiable function?' (did not define)
(78) What is '$E_n$?'
(78) Proof.
(83) Since ~ such that , (remove `therefore')
(89) reove `hence'
(90; 91; 129) remove `:'
(96) computational
(116) Figures 1 and 2.
(130) remove `thus'
(132) proof, since the error equation
(143; 145), Figures 3 and 4
(*) unify the references, Refs [3]/[4], Bull. Austral. Math. Soc./Mathematics,
(198) nonlinear
(230) Volume (?), Pages (?)
(*) unify the references, Refs [21]/[27], Computers &/Mathematics and
(255), pages (?)
The present form is not readable.
The authors did not read the final version before submitting the paper.
Author Response
The file is attached.

Round 2
Reviewer 1 Report
it can be published,the only suggestion is to chech the language again.
Author Response
The file is attached.

Reviewer 3 Report
In this work, the authors introduce a new class of frozen Jacobian multi-step iterative methods to solve systems of nonlinear equations and numerical examples are provided to show the proposed algorithms are very useful.
The results are interesting and new. But there are some errors in typos and grammar. For example,
(1) P.1(-14), the relation is not clear between x and $x$
(2) P.3(-11; -8), remove `”’
(3) P.3(-8), `for more details on ~ therein’ should be located outside the definition.
(4) P.3(-1), Suppose that ~ . Using Taylor’s expansion [31], we get
(5) P.4(+6), `.’ should be replaced by `,’
(6) P.4(+7), where ~ matrix whose order is the same as the order
(7) P.4(-12), (6). Similarly, the Taylor’s
(8) P.4(-2), Finally, since
(9) P.5(+1), we have
(10) P.5(-18), which was defined
(11) P.7(+1; +2), What is JFA? (did not define)
(12) Refs [5], [7], [27] Int. J. Comput. Math.,
(13) Ref [8] Phys. A:
(14) Ref [10] Nonlinear Anal, {\bf 62} (2005), no. 1, 179-194.
(15) Ref [14] J. Interdis. Math.,
(16) Ref [17] J. Comput. Appl. Math.,
(17) Ref [19] Int. J. Comput. Sci. Math.,
(18) Refs [20], [26] Appl. Math. Comput.
(19) Refs [22], [28] Comput. Math. Appl.,
(20) Ref [23] J. Indust. Management Optim.,
(21) Ref [30] Aian J. Math. Sci.,
(22) Ref [31] Soc. Indust. Appl. Math., Voulme (?), pages (??)
(23) Ref [32] J. Comput. Appl. Math.,
(24) Compare Refs [29] and [33]
(25) Ref [35] Int. J. Pure Appl. Math.,
(26) Ref [36] J. Optim. Theory Appl.,
(27) Ref [37] J. Comput. Appl. Math.,
(28) Ref [38] Comput. Math. Appl.,
(29) Ref [40] Appl. Math. Comput.
(30) Ref [41] Math. Comput. Modell.
(31) Ref [43] Appl. Math. Comput.
(32) Ref [44] Int. J. Comput. Appl.
The authors have to read the final revised version before submitting the revised version.
Author Response
The file is attached.
